# Chitooligosaccharide from Pacific White Shrimp Shell Chitosan Ameliorates Inflammation and Oxidative Stress via NF-κB, Erk1/2, Akt and Nrf2/HO-1 Pathways in LPS-Induced RAW264.7 Macrophage Cells

**DOI:** 10.3390/foods12142740

**Published:** 2023-07-19

**Authors:** Lalita Chotphruethipong, Pithi Chanvorachote, Ratchaneekorn Reudhabibadh, Avtar Singh, Soottawat Benjakul, Sittiruk Roytrakul, Pilaiwanwadee Hutamekalin

**Affiliations:** 1Department of Food Science, Faculty of Science, Burapha University, Mueang Chonburi, Chonburi 20131, Thailand; lalita.ch@go.buu.ac.th; 2Department of Pharmacology and Physiology, Faculty of Pharmaceutical Sciences, Chulalongkorn University, Bangkok 10330, Thailand; pithi.c@chula.c.th; 3Center of Excellence in Cancer Cell and Molecular Biology, Faculty of Pharmaceutical Sciences, Chulalongkorn University, Bangkok 10330, Thailand; 4Faculty of Science and Technology, Hatyai University, Hat Yai 90110, Thailand; ratchaneekorn.re@hu.ac.th; 5International Center of Excellence in Seafood Science and Innovation, Faculty of Agro-Industry, Prince of Songkla University, Hat Yai 90110, Thailand; avtar.s@psu.ac.th (A.S.); soottawat.b@psu.ac.th (S.B.); 6Functional Ingredients and Food Innovation Research Group, National Center for Genetic Engineering and Biotechnology, National Science and Technology Development Agency, Pathum Thani 12120, Thailand; sittiruk@biotec.or.th; 7Division of Health and Applied Sciences, Faculty of Science, Prince of Songkla University, Hat Yai 90110, Thailand

**Keywords:** chitooligosaccharide, shrimp shell, anti-inflammatory, cell signaling pathways, macrophage cells, antioxidant

## Abstract

Chitooligosaccharide (COS), found in both insects and marine sources, has several bioactivities, such as anti-inflammation and antioxidant activities. However, the mechanism of shrimp shell COS on retardation of inflammatory and antioxidant effects is limited. Therefore, the aim of this study is to examine the mechanism of the aforementioned activities of COS in LPS-activated RAW264.7 macrophage cells. COS significantly improved cell viability in LPS-activated cells. COS at the level of 500 µg/mL could reduce the TNF-α, NO and IL-6 generations in LPS-activated cells (*p* < 0.05). Furthermore, COS could reduce ROS formation, NF-κB overactivation, phosphorylation of Erk1/2 and Akt and Nrf2/HO-1 in LPS-exposed cells. These results indicate that COS manifests anti-inflammatory activity and antioxidant action via NF-κB, Erk1/2, Akt and Nrf2/HO-1 signaling with an increasing relevance for inflammatory disorders.

## 1. Introduction

Inflammation is the bodily reaction in preventing tissue injury governed by several results, such as bacteria and chemical agents [1]. Lipopolysaccharide (LPS), the endotoxin, is widely used in investigations of the mechanism of inflammatory responses [2,3]. LPS activates macrophages through toll-like receptor 4 (TLR4), which further regulates MAPK signaling and NF-κB [4,5]. Subsequently, the response of TLR4 to LPS leads to the production of tumor necrosis factor (TNF-α), nitric oxide (NO) and interleukin (IL)-6 [6,7]. Several studies indicated that suppression of LPS inhibits the production of inflammatory mediators [8,9]. In addition, activation of macrophages also promotes reactive oxygen species (ROS) generation, which causes the risk of chronic diseases [10]. A high level of ROS can modulate NF-κB and Nrf2 signaling [11]. Under the production of oxidative stress, Nrf2 regulates HO-1 in LPS-challenged RAW264.7 activation [12,13]. Additionally, LPS-induced ROS production was attenuated by modulating nuclear translocation of Nrf2 via the PI3K/Akt signaling pathway [14]. Thus, the controlling liberation of those mediators activated by macrophages is necessary. Generally, antioxidants can protect against cell damage caused by ROS or the inflammatory mediators [15,16,17].

Chitooligosaccharide (COS) is the hydrolyzed result of chitosan, which is easily soluble in water [18]. Commonly, COS has been found in several sources, including insect and marine, etc. [19]. Shrimp shell chitosan is a potential marine resource with various bioactivities [20]. However, the COS preparation process is important in determining its characteristics and bioactivity. The enzymatic method has been extensively used for preparing COS. Nevertheless, this method has some limitations, such as high cost. Also, chitosan with high viscosity can be obtained after non-specific enzyme hydrolysis, leading to a low degree of hydrolysis [21]. Mittal et al. [20] revealed that COS prepared using the OH-H_2_O_2_ method had high antioxidant and antimicrobial activities, especially when 1% H_2_O_2_ was used. Varieties of COS production showed that commercial COS had some ability to inhibit IL-6 and TNF-α productions of LPS-activated RAW264.7 macrophage and possessed high anti-inflammatory activity in BALB/c mice [22,23]. Although other sources also have the anti-inflammation and antioxidant functions, no information regarding the mechanism of COS from shrimp shell chitosan on anti-inflammatory activities and oxidative stress exists [24,25]. Thus, this study aims to investigate the influence of COS at various levels on the viability of LPS-activated RAW264.7 cells. Moreover, its ability in obstructing pro-inflammatory cytokines and the related pathways was also examined.

## 2. Materials and Methods

### 2.1. Chemicals and Material

RAW264.7 cells were obtained from ATCC (Bethesda, MD, USA). Griess reagent, ELISA kit and MTT were procured from Sigma-Aldrich (St. Louis, MO, USA). Fetal bovine serum and RPMI 1640 were supplied by Gibco^®^, Thermo Fisher Scientific, Inc. (Waltham, MA, USA). Chitosan (*M*_W_: ~2.1 × 10^3^ kDa and degree of deacetylation: 82%) was given by Marine Bio Resources Co., Ltd., Samutsakhon, Thailand. All the other chemicals and reagents were analytical grade and obtained from commercial sources.

### 2.2. Chitooligosaccharide (COS) Production

COS was processed using the oxidative hydrolysis method with the aid of hydrogen peroxide (H_2_O_2_) [20]; 1% chitosan (CS) was initially dissolved in 2% acetic acid overnight, followed by pH adjustment to 5.5. Thereafter, H_2_O_2_ was mixed by CS solution and heated at 60 °C for 2 h. The mixtures were cooled in iced water and pH adjusted to 7. The undissolved matter was removed by centrifugation and the supernatant was pooled. The COS solution was lyophilized, and the COS powders were kept at −40 °C until used. The preparation process of COS is shown in Figure 1. Degree of polymerization (DP) of COS and *M*_W_ were determined using a MALDI-TOF-MS and gel permeation chromatography techniques, respectively [20]. DP of COS was 3–6.

### 2.3. Cell Viability Test

RAW264.7 cells (1 × 10^4^ cells/well) were seeded in a 96-well plate. Thereafter, LPS at 0.001–10 µg/mL or COS at 25–1000 µg/mL were added to the cells. Subsequently, cell viability was estimated using an MTT assay [17]. Results were determined as percentages in comparison to the control values. The levels of LPS or COS without cytotoxicity were selected for further experiments.

To investigate the effectiveness of COS towards LPS-activated cells, the cells were treated with COS at the selected levels (25–500 µg/mL) for 24 h, followed by adding 1 µg/mL LPS for another 24 h. After adding LPS, cytotoxicity was examined [17]. Apoptotic cells were stained with Hoechst 33,342 dye and the images were captured using a fluorescence microscope [16].

### 2.4. Inflammatory Cytokine Assay

COS at 500 µg/mL was added to cells and kept in a cell incubator for 24 h [16]. After incubation, 1 µg/mL LPS was added to cells for another 24 h. Subsequently, the supernatant was pooled and used to measure TNF-α and IL-6 inflammatory cytokines by ELISA kit. The nitrate level was examined by Griess reagent. The protocols were conducted as per the manufacturer’s instructions.

### 2.5. Intracellular ROS Measurement

The intracellular ROS level was tested by staining with DCFH-DA dye. Cells (5 × 10^4^ cells/well) were pre-incubated with COS and subsequently added LPS for 24 h. The cells were probed with dye for 1 h, as detailed by Chotphruethipong et al. [16]. Fluorescence intensity was calculated by comparing to the control at the same time. Fluorescence intensity was measured at 485 nm and 530 nm for excitation and emission, respectively.

### 2.6. Western Blot Analysis

Cells (1 × 10^6^ cells/dish) were seeded into a 35 mm dish for 24 h and subsequently 500 μg/mL COS was added for 24 h with or without an antagonist for 1 h, and treated with LPS (1 µg/mL) for 30 min. The cells were lysed and the protein content was measured [26]. The proteins were loaded on SDS-PAGE and transferred onto PVDF membranes. BSA (3%, *w*/*v*) or skimmed milk (5%, *w*/*v*) was used to block the membranes before incubating with primary antibodies against caspase-3, cleaved caspase-3, Akt, p-Akt, Erk1/2 and p-Erk1/2 for overnight. The secondary antibody grafted with HRP was incubated for 2 h. The protein signals were detected with chemiluminescence substrate. Loading control used was β-actin.

### 2.7. Statistical Analysis

Four independent experiments were examined. The data were reported as the average ± standard deviation (SD). The significant difference between groups was done by one-way ANOVA with Tukey’s post hoc test.

## 3. Results

### 3.1. Impact of COS or LPS at Various Concentrations on Viability of RAW264.7 Macrophage Cell

To investigate the protective effect of COS, LPS was applied to trigger cytotoxicity in RAW264.7 cells. The impact of COS on cell viability is presented in Figure 2a. Treatment with 25–500 µg/mL COS had no cytotoxicity on RAW264.7 cells as compared to control (without COS treatment) (*p* > 0.05), whereas a high level of COS (1000 µg/mL) resulted in reduced cell viability (*p* < 0.05). Since cell viability was similar between COS-treated cells at 25–500 µg/mL and control, these levels were chosen for further study. When LPS at different concentrations (0.001–10 µg/mL) was used to assess cell viability (Figure 2b), high levels of LPS (1–10 µg/mL) reduced cell viability greater than control (*p* < 0.05). Conversely, LPS-treated cells at 10 µg/mL (75.08%) had lower cell viability than LPS-treated cell at 1 µg/mL (84.73%). Therefore, LPS at the level of 1 µg/mL was selected for the subsequent trials. To verify whether COS could diminish LPS-induced cytotoxicity, cells were pre-incubated with COS at 25–500 µg/mL for 24 h before incubation with LPS at 1 µg/mL for 24 h. As depicted in Figure 2c, cells added with 1 µg/mL LPS had lowered viability compared to the untreated group (control) (*p* < 0.05). Nevertheless, pretreatment with COS at high concentrations (100–500 µg/mL) could inhibit LPS-induced cytotoxicity (*p* < 0.05). The highest inhibitory effect was found for cells treated with COS at 500 µg/mL. Thus, COS at the concentration of 500 µg/mL was an appropriate level for further investigations.

### 3.2. Effect of COS Combined with LPS at Selected Concentrations on Apoptotic Cells, Caspase 3 and Cleaved-Caspase 3 Expressions in LPS-Treated Cells

It is well known that LPS is a potential inducer of cell death [27,28]. Thus, the effect of COS-attenuated LPS-induced cell death was determined. Cell apoptosis was tested by staining with Hoechst 33,342 and western blot analysis. The finding showed that pretreatment with 500 μg/mL COS prior to LPS treatment could reduce death of LPS-treated cells excellently as evidenced by the lowered number of dead cells, compared with LPS alone. A significant presence of DNA fragmentation (indicated as arrows in Figure 3a). The percentage of apoptotic cells was then quantified (Figure 3b). Correspondingly, western blot results also confirmed that pretreating the cells with COS prior to LPS treatment could prevent death of cells, as witnessed by the decreased expression of cleaved caspase 3, compared to that of LPS-activated cells alone (*p* < 0.05) (Figure 3c). The expression level of cleaved caspase-3/caspase-3 was quantified (Figure 3d). These investigations support the view that COS protect LPS-induced cytotoxicity by inhibition of apoptotic proteins.

### 3.3. Protective Effect of COS against Inflammation

Several lines of study showed that LPS-stimulated pro-inflammatory cytokines, NO and NF-*κ*B [29,30]. In the present study, the anti-inflammatory activity of COS towards LPS-activated cell was tested by detection of IL-6, NO and TNF-α levels. As illustrated in Figure 4a, LPS-activated cells had the highest NO level compared to control (*p* < 0.05). With pretreating COS into LPS-induced cells, a decrease in the NO level was observed. Also, the NO level of LPS-induced cells pretreated with COS was similar to control (*p* > 0.05), demonstrating that COS had excellent anti-inflammatory activity. When considering TNF-α (Figure 4b) and IL-6 levels (Figure 4c) of all samples tested, the results were associated with NO inhibitory activity. The lowered levels of IL-6 and TNF-α after pretreating COS reflected effectiveness against mediator release, as affected by LPS treatment. Apart from those aforementioned results, western blot analysis also confirmed that COS had inflammatory activity, as evidenced by a lower level of NF-*κ*B expression than LPS-treated cells (*p* < 0.05) (Figure 4d).

### 3.4. Effect of COS at Selected Concentration on ROS Prohibition of LPS-Activated RAW264.7 Cells

A previous study demonstrated that LPS enhanced ROS production [31]. Thus, the effect of COS-inhibited intracellular ROS generation as governed by LPS was examined. The ability of COS to inhibit intracellular ROS of LPS-treated cells at various incubation times (1–24 h) is depicted in Figure 5. The results present a significant increase in the intracellular ROS level in the LPS-treated group compared with the control group at 3 h (*p* < 0.05) while COS or NAC (ROS scavenger) alone did not result in any significant alteration in ROS production. In contrast, pretreatment with COS and NAC dramatically decreased the intracellular ROS formation in the LPS-treated group. Results suggested that COS could be used for protecting LPS-induced ROS generation in cells.

### 3.5. Effect of COS on Inhibition of Nrf-2 and HO-1 in LPS-Activated RAW264.7 Cells

Previous study has reported that Nrf-2 and HO-1 play an essential role in regulating ROS-stimulated cell injury in inflammatory cells [32]. To study whether Nrf-2 and HO-1 are associated with the LPS-stimulated inflammatory response, RAW264.7 macrophage cells were observed by western blot analysis. The cells treated with LPS alone were found to have the marked productions in Nrf2 and HO-1, compared to the control group (*p* < 0.05) (Figure 6a–d). Decreased expression levels of both proteins were found for COS-pretreated cells before exposure to LPS treatment (*p* < 0.05). These data indicate that COS can protect LPS-induced RAW264.7 cells. Taken together, the results reveal that COS has potential in the prevention of inflammation and ROS production as formed by the LPS involved with the Nrf-2 and HO-1 signaling pathway.

### 3.6. Effect of COS on Inhibition of Erk1/2 and Akt Expressions in LPS-Activated RAW264.7 Cells

A recent study reported that LPS activated Erk and Akt signaling pathway during inflammation [33]. To investigate the basic mechanism of COS in cell protection from LPS-stimulated inflammation, the expressions of Erk1/2 and Akt were examined. The expression of Erk1/2 and Akt levels was increased in LPS-activated cells (*p* < 0.05), while cells pretreated with COS before exposure to LPS were decreased (*p* < 0.05) (Figure 7). To verify whether COS gave a protective effect via Erk1/2 and Akt, PD98095 and perifosine were applied as Erk1/2 and Akt inhibitors, respectively. It was noted that perifosine could induce the expression levels of Akt and Erk1/2 to be higher than LPS-activated cells (*p* < 0.05), reflecting that COS had an ameliorating effect on LPS-activated inflammation associated with downregulating Erk1/2 and Akt.

## 4. Discussion

LPS is a potent stimulator for macrophages to release pro-inflammatory cytokines such as IL-6 and TNF-α, NO and NF-*κ*B [27]. Accumulating evidence demonstrates that ROS is also associated with inflammation [26,27]. Increasing agents have been equipped to suppress inflammation [17,34]. COS is one of the promising agents that represents the anti-inflammatory process. However, the mechanism of COS is still elusive. Therefore, the present study aims to investigate the mechanism of COS from shrimp shell in LPS-activated RAW264.6 macrophage cells. Our results indicate that applied COS had no cytotoxic effect on RAW264.7 cells at levels ranging from 25 to 500 µg/mL. Higher levels (1000 µg/mL) led to decreased cell viability. In general, COS from Pacific white shrimp shell chitosan was able to scavenge free radicals generated [20]. Nevertheless, an excessive level of COS likely augmented intracellular oxidant levels, resulting in cell death. The data were related with Ei et al. [35], who reported that the use of COS at levels higher than 500 μg/mL decreased the survival of human EA.hy926 endothelial cells. Thus, the level of COS was a significant factor affecting cell viability, in which the proper level of COS in the present study was at 500 µg/mL. Apart from the COS level, the level of LPS also affected cell viability, in which the maximum level used was 1 μg/mL.

When the protective effect COS on the apoptotic pathway of RAW264.7 cells was studied, COS could protect from damage to cells as influenced by LPS activation. Anil [36] revealed that COS possessed great bioactivities, particularly antioxidant and anti-inflammatory activities, owing to the existence of several active functional groups, including a protonated amino group at C-2 position and hydroxyl groups at C-3 and C-6 positions. Those functional groups could scavenge free radicals in cells. In general, antioxidant activity (AA) of COS depends on DP [37]. Mengibar et al. [38] reported that COS with low DP (3–7) exhibited high AA. COS from Pacific white shrimp shell chitosan had low DP (3–6), in which possibly showed the powerful AA, especially radical quenching activity. This result conformed to the apoptotic cell and expression level of cleaved caspase 3 in LPS-activated RAW264.7 cells, in which COS could reduce cell death. Cleaved caspase 3 is commonly used to determine the apoptotic pathway [39]. Thus, the lowered level of cleaved caspase 3 in COS-treated cells before induction by LPS indicated the ability of COS to prevent LPS-treated macrophage cells.

When RAW264.7 cells were acclimatized to assess the anti-inflammatory activity of COS, LPS at the selected level initiates inflammation in RAW264.7 cells by activating mediators of the inflammatory response by an increase in IL-6, NO and TNF-α protein expressions. COS could arrest LPS-generated inflammation by decreasing the expression of NO and TNF-α and IL-6 proteins. These results are similar to a previous finding that COS could prevent LPS-activated inflammation [40]. Macrophages are widely used to investigate the effectiveness of bioactive compounds against inflammation [41,42]. Several studies demonstrated that LPS induced inflammatory responses in macrophage cells which contribute to the formation of pro-inflammatory mediators [3,43,44]. Moreover, the expression level of NF-*κ*B also indicated the ability of bioactive compounds to prevent inflammation [45]. The NF-*κ*B activation signaling pathway generates IL-6 and TNF-α proinflammatory cytokines [46,47,48]. A lowered expression level of NF-*κ*B was perceived in COS-pretreated cells before adding LPS. This finding reflected that COS had an inflammatory activity towards LPS-activated cells.

Additionally, the effectiveness of COS on ROS inhibition of LPS-activated RAW264.7 cells was investigated. In general, ROS generation plays a crucial role in various aspects; however, excessive level of ROS can induce the pathogenesis of many diseases [49]. It was noted that ROS generation leads to the onset of inflammation by induction of NF-*κ*B and pro-inflammatory cytokines, namely TNF-α, IL-6 [8]. Consistent with this, NF-*κ*B, TNF-α and IL-6 levels in this study appeared to increase in response to ROS treatment, whereas the decreased levels of NF-*κ*B, TNF-α and IL-6 activated by LPS treatment were found in the presence of COS. In addition, our data revealed that antioxidant agent NAC suppressed NF-*κ*B levels. Several lines of evidence indicate that Nrf2 is a crucial molecular mechanism of oxidative stress and inflammation [50]. Nrf2 further triggers antioxidant enzyme HO-1. This study indicates that COS can restore the expression level of Nrf2/HO-1 in LPS-activated macrophage cells. Collectively, these suggest that ROS production mediated COS-diminished inflammation via Nrf2/HO-1.

When RAW264.7 cells were used to test the influence of COS on controlling of Erk1/2 and Akt signaling pathways, COS exerts anti-inflammation associated with Erk1/2 and Akt signaling pathways and contributes to releasing TNF-α, NO and IL-6 inflammatory mediators in RAW264.7 cells activated by LPS. Also, COS could alleviate LPS-activated oxidative stress through Nrf2/HO-1 signaling. Thus, COS had a potent effect on inhibiting oxidative stress and inflammation in macrophage cells induced by LPS. Several studies demonstrated that the Akt, NF-*κ*B and MAPK signaling pathways take part in inflammation [14,51]. Our findings agree with a previous study which COS decreased the expression level of p-Akt/Akt and p-Erk1/2/Erk1/2 in LPS-treated cells. In addition, our data go along with the results of the previous study that Akt also acts as a key signaling pathway for cell survival by inhibiting macrophage cells death in COS-treated cells. [52]. Thus, COS had an anti-inflammatory and antioxidative stress effect on LPS-induced macrophage cells via the Erk1/2, Akt, NF-*κ*B and Nrf2/HO-1 pathways, in which COS might be used for a promising functional compound for the ministration of inflammatory disease.

## 5. Conclusions

COS from Pacific white shrimp shell chitosan had no cytotoxicity on RAW264.7 cells when levels lower than 1000 µg/mL were used. The use of COS at the level of 500 µg/mL could protect against the death of LPS-induced cells better than other levels. COS at 500 µg/mL had the ability to fight inflammation by inhibiting the expression of NO, NF-*κ*B, IL-6 and TNF-α proteins in LPS-induced cells. Moreover, it could inhibit ROS generation in cells via the Nrf2/HO-1 pathway. Also, COS decreased the expression of p-Erk1/2/Erk1/2 and p-Akt/Akt proteins in LPS-treated cells related to cell survival. Therefore, COS can be a functional ingredient with bioactivities, especially antioxidant and anti-inflammatory activities.

## Figures and Tables

**Figure 1 foods-12-02740-f001:**
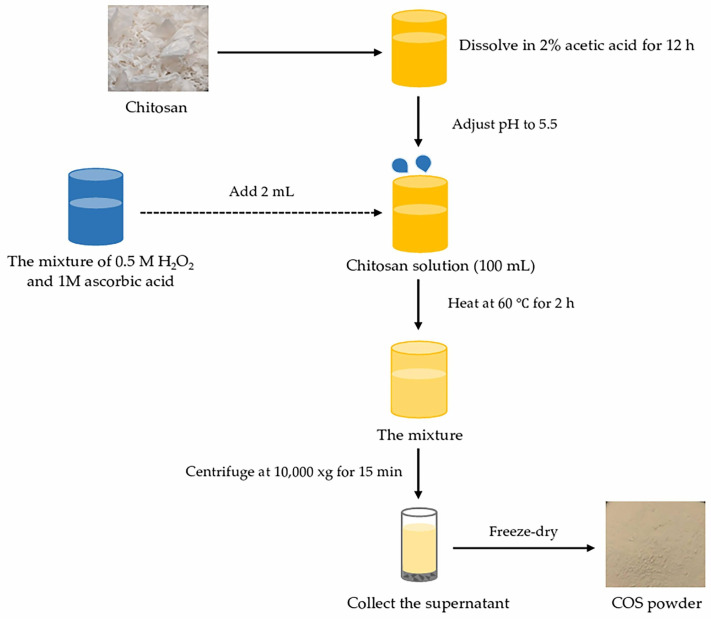
Preparation process of chitooligosaccharide (COS) from Pacific white shrimp shell chitosan using oxidative hydrolysis method.

**Figure 2 foods-12-02740-f002:**
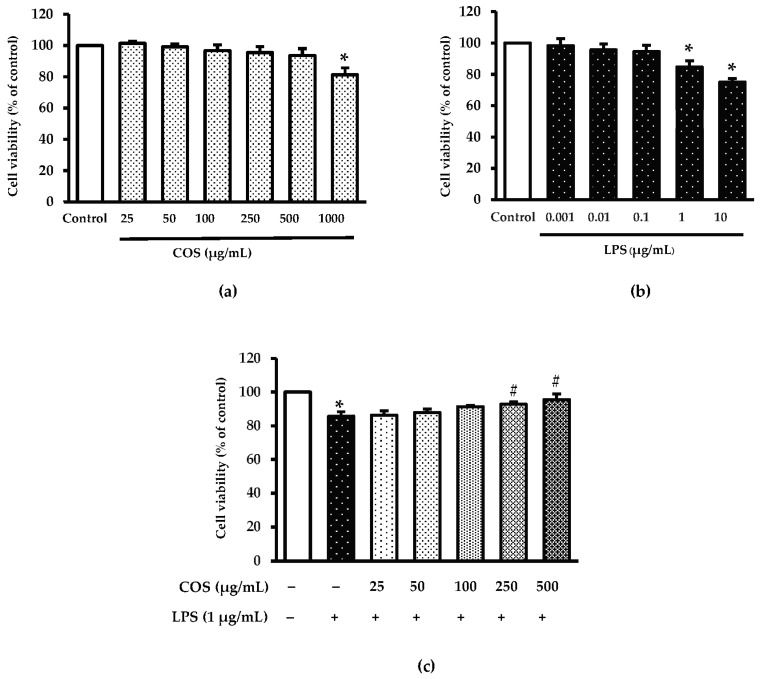
Chitooligosaccharide (COS) prevented lipopolysaccharide (LPS)-induced cytotoxicity in RAW264.7 macrophage cells. (**a**) The cells were treated various concentrations of COS (25–1000 μg/mL) for 24 h. (**b**) The cells were treated with various concentrations of LPS (0.001–10 μg/mL) for 24 h. (**c**) The cells were pretreated with COS (25–500 μg/mL) for 24 h followed by 1 μg/mL LPS for 24 h. Cell viability was measured by 0.5 mg/mL of MTT assay. Results are shown as the mean ± SD (*n* = 4). * *p* < 0.05 vs. control cells. # *p* < 0.05 vs. the LPS group.

**Figure 3 foods-12-02740-f003:**
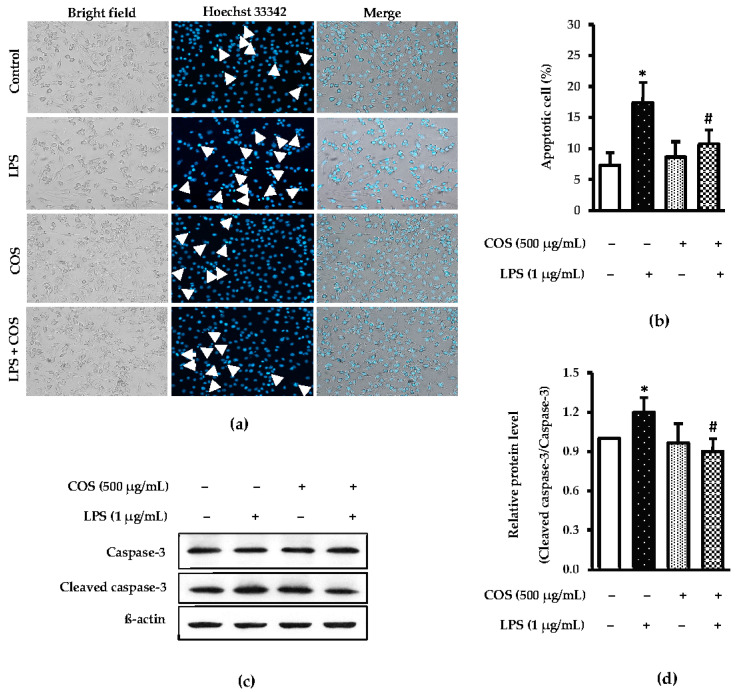
COS ameliorated LPS-induced apoptosis in RAW264.7 cells. Cells were pretreated with 500 μg/mL COS for 24 h, followed by 1 μg/mL LPS for 24 h. Next, the cells were examined by Hoechst 33342 staining and western blotting. (**a**) The cells stained with Hoechst 33342. Nuclear fragmentations were evaluated under fluorescence microscope (20×). DNA fragmentations were indicated as arrows. (**b**) Bar graph represents percentage of apoptotic cells. (**c**) Expression of caspase-3 and cleaved caspase-3 were evaluated by western blot analysis. (**d**) Bar graph shows quantification of cleaved caspase-3/caspase-3. Data were normalized by using β-actin as control. Results are shown as the mean ± SD (*n* = 4). * *p* < 0.05 vs. control cells. # *p* < 0.05 vs. the LPS group.

**Figure 4 foods-12-02740-f004:**
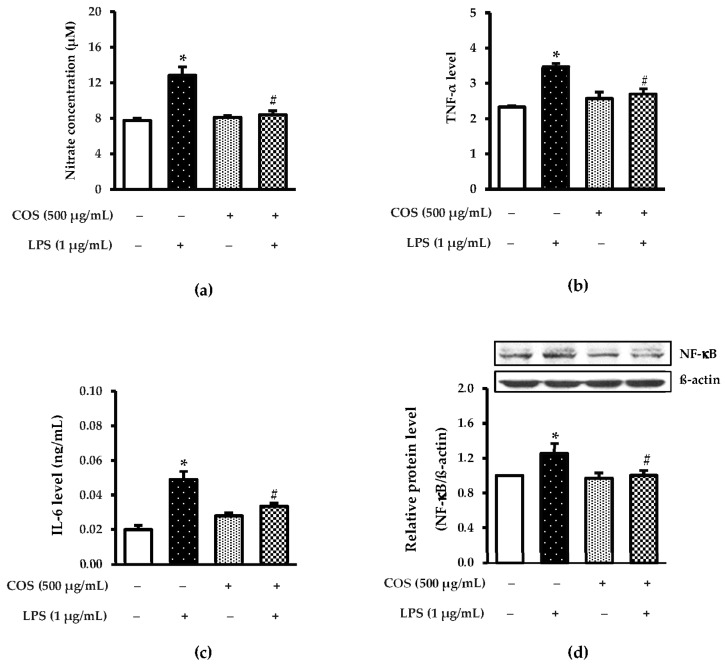
COS alleviated pro-inflammatory cytokines, nitric oxide (NO) and NF-*k*B in LPS-activated RAW264.7 cells. The cells were pretreated with 500 μg/mL COS for 24 h, followed by 1 μg/mL LPS for 24 h. Level of NO, TNF-α, and IL-6 was examined by using ELISA assay. Supernatants of conditioned media were collected for (**a**) nitrite; (**b**) TNF-α; (**c**) IL-6 measurement. (**d**) Expression of NF-*κ*B was determined by western blotting. Bar graph shows quantification of NF-*κ*B to β-actin. Results are shown as the mean ± SD (*n* = 4). * *p* < 0.05 vs. control cells. # *p* < 0.05 vs. the LPS group.

**Figure 5 foods-12-02740-f005:**
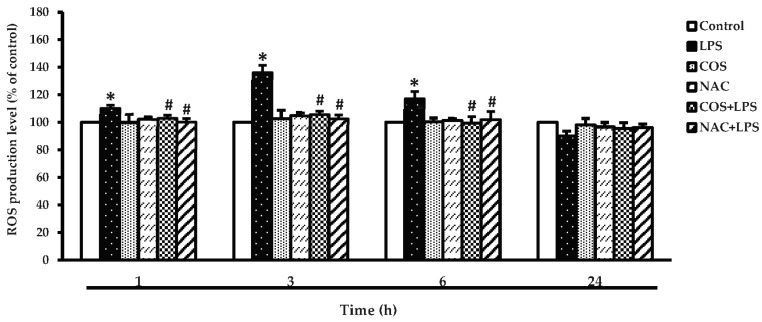
COS prevented LPS-induced intracellular reactive oxygen species (ROS) production in RAW264.7 cells. The cells were treated with 1 μg/mL LPS for 24 h in the presence or absence of 500 μg/mL COS for 24 h. In addition, the cells were treated with 1 μg/mL LPS for 24 h in the presence or absence of 1 mM ROS scavenger (NAC) for 1 h. DCFH-DA was used for detecting ROS generation. Results are shown as the mean ± SD (*n* = 4). * *p* < 0.05 vs. control cells. # *p* < 0.05 vs. the LPS group.

**Figure 6 foods-12-02740-f006:**
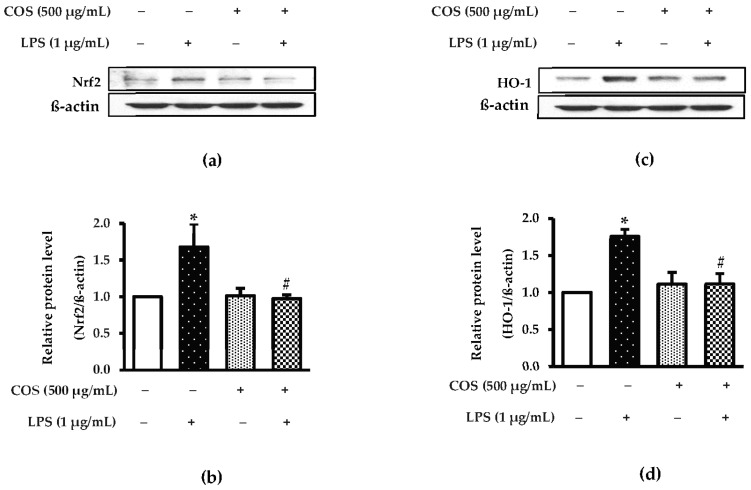
COS diminished LPS-activated Nrf2/HO-1 in RAW264.7 cells. The cells were preincubated with 500 μg/mL COS for 24 h, followed by 1 μg/mL LPS for 24 h. The expression level of Nrf2/HO-1 was investigated using western blot analysis. (**a**) Expression of Nrf2. (**b**) Quantification of Nrf2. (**c**) Expression of HO-1. (**d**) Quantification of HO-1. The signal intensity was normalized to the corresponding intensity of β-actin. Results are shown as the mean ± SD (*n* = 4). * *p* < 0.05 vs. control cells. # *p* < 0.05 vs. the LPS group.

**Figure 7 foods-12-02740-f007:**
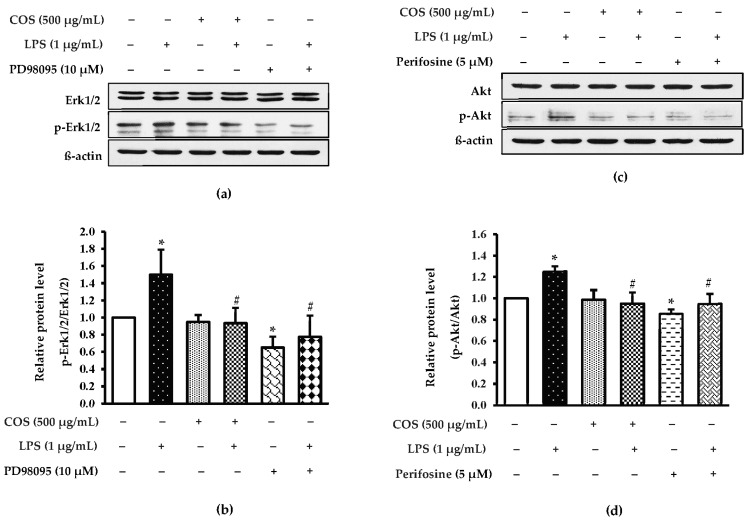
COS ameliorated LPS-activated Erk1/2 and Akt induction in RAW264.7 cells. The cells were pretreated with 500 μg/mL COS for 24 h, followed by with or without the presence of signaling molecule inhibitor (10 μM PD98095 Erk1/2 inhibitor, 5 μM Perifosine Akt inhibitor) and 1 μg/mL LPS 24 h. Protein expressions were performed by western blot analysis. (**a**) Expression of Erk1/2 and p-Erk1/2. (**b**) Quantification of p-Erk1/2/Erk1/2. (**c**) Expression of Akt. (**d**) Quantification of pAkt/Akt. Quantification of the proteins is reported as the relative of the proteins to beta actin. Results are shown as the mean ± SD (*n* = 4). * *p* < 0.05 vs. control cells. # *p* < 0.05 vs. the LPS group.

## Data Availability

The data presented in this study are available on request from the corresponding author.

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
