# Peer review of "Chitooligosaccharide from Pacific White Shrimp Shell Chitosan Ameliorates Inflammation and Oxidative Stress via NF-κB, Erk1/2, Akt and Nrf2/HO-1 Pathways in LPS-Induced RAW264.7 Macrophage Cells"

_foods, 2023, doi:10.3390/foods12142740_

Round 1
Reviewer 1 Report
The manuscript is well written. However, it needs minor revison. All suggestions were indicated in the manuscript file.

Author Response
Dear reviewer,
We have edited the manuscript and answered the questions according to the reviewer's comments. The revision and rebuttal as shown below:
The manuscript is well written. However, it needs minor revison. All suggestions were indicated in the manuscript file.
******Thank you for the invaluable comments. All queries have been responded and the corrections have been made as per the reviewer’s suggestion as highlighted in yellow.
Line 45-48: In this part of the text, it should be also added that: antioxidants protect cells against oxidative damage. The novel references should be added:
Ramazani, N., Mahd Gharebagh, F., Soleimanzadeh, A., Arslan, H. O., Keles, E., Gradinarska‐Yanakieva, D. G., ... & Dinç, D. A. (2023). The influence of L‐proline and fulvic acid on oxidative stress and semen quality of buffalo bull semen following cryopreservation. Veterinary Medicine and Science. Also it should be mentioned and supported with novel & accurate references that: oxidative stress plays an essential role in the emergence of a number of chronic disorders such as diabetes and cancer by inducing inflammation.
According to the Acaroz, U., Ince, S., Arslan-Acaroz, D., Gurler, Z., Demirel, H. H., Kucukkurt, I., ... & Zhu, K. (2019). Bisphenol-A induced oxidative stress, inflammatory gene expression, and metabolic and histopathological changes in male Wistar albino rats: protective role of boron. Toxicology research, 8(2), 262-269.
This reference should be added.
******The additional information has been provided in text. Also, the suggested references have been added (Please see line 46-47, 51-52 and reference list). We cannot provide all sentences as pre the reviewer’s suggestion due to high similarity index. Sorry for this.
Line 69-73: It should be added here that: All the other chemicals and reagents were of analytical reagent grade purchased from commercial sources.
******The suggestion sentence have been stated in text (please see line 74-75, page 2). Thank you.
All figures should also be given as table.
******The values presented in graphs are cell viability. Nevertheless, treatments tested in each graph are different. Form Fig.1a, data on the y-axis, it is cell viability as affected by COS at different levels. The data shown in Fig.1b represents cell viability after treatment with LPS at various levels. For Fig. 1c, it is combined effect of COS and LPS at the selected levels without cytotoxicity, in which the selected data are obtained from Fig.1a-b. Thus, to better understanding for reader, we decided to present the data as graphs. Several studies also presented their data in the similar way (Panyathep et al.; Tong et al.; Shkryl et al.). Thank you for comment.
References
Panyathep, A., Punturee, K., Chewonarin. Inhibitory Effects of Chlorogenic Acid Containing Green Coffee Bean Extract on Lipopolysaccharide-Induced Inflammatory Responses and Progression of Colon Cancer Cell Line. Foods 2023, 12(14), 2648; https://doi.org/10.3390/foods12142648
Tong, W., Chen, X., Song, X., Chen, Y., Jia, R., Zou, Y., Li, L., Yin, L., He, C., Liang, X., Ye, G., Lv, C., Lin, J., Yin, Z. Resveratrol inhibits LPS induced inflammation through suppressing the signaling cascades of TLR4 NF κB/MAPKs/IRF3. Experimental And Therapeutic Medicine 2020, 19: 1824-1834.
Shkryl, Y.N., Tchernoded ,G.K., Yugay, Y.A., Grigorchuk, V.P., Sorokina, M.R., Gorpenchenko, T.Y., Kudinova, O.D., Degtyarenko, A.I., Onishchenko, M.S., Shved, N.A., Kumeiko, V.V., Bulgakov, V.P. Enhanced Production of Nitrogenated Metabolites with Anticancer Potential in Aristolochia manshuriensis Hairy Root Cultures. International Journal of Molecular Sciences 2023, 24(14), 11240; https://doi.org/10.3390/ijms241411240
Also, we have attached the revised manuscript in the box. Please see the attachment.

Reviewer 2 Report
In this paper, the mechanism of anti-inflammation and antioxidant of COS in LPS stimulated RAW264.7 macrophage cells was determined. It was found that COS significantly improved cell viability in LPS-treated. Furthermore, COS could diminish ROS formation, NF-kB over activation, phosphorylation of Erk1/2 and Akt and Nrf2/HO-1 in LPS-exposed cells. It is much interesting and minor comments are following:
1. What are the degree of polymerization and polymerization degree distribution of COS. Moreover, how many is the yield of COS from chitosan.
2. Line 74, chitosan was given by Marine Bio Resources Co., Ltd, is it related with Pacific White Shrimp Shell Chitosan in title? Why did you emphasize the Pacific White Shrimp Shell Chitosan? Does COS from others have not the anti-inflammation and antioxidant functions?
3. Line 73, italic M and subscript w for Mw. Line 80, italic for “g”. Lines 96 and 102, 5 × 104 (not x). Line 108, 35 mm. Line 128, P-value.
Author Response
Dear reviewer,
We have edited the manuscript and answered the questions according to the reviewer's comments. The revision and rebuttal are shown below:
Comments and Suggestions for Authors
In this paper, the mechanism of anti-inflammation and antioxidant of COS in LPS stimulated RAW264.7 macrophage cells was determined. It was found that COS significantly improved cell viability in LPS-treated. Furthermore, COS could diminish ROS formation, NF-kB over activation, phosphorylation of Erk1/2 and Akt and Nrf2/HO-1 in LPS-exposed cells. It is much interesting and minor comments are following:
******Thank you for understanding our work. All queries have been responded and the corrections have been made as per the reviewer’s suggestion as highlighted in green.
What are the degree of polymerization and polymerization degree distribution of COS?. Moreover, how many is the yield of COS from chitosan?.
******The degree of polymerisation (DP) of COS was measured by matrix-assisted laser desorption/ionisation-time of flight-mass spectrometry (MALDI-TOF-MS) (Mittal et al., 2022). Trimer with an acetyl group was computed as follows: 179 (C6H13O5N) + 2 × 161 (C6H11O4N) + 2 × 43 (C2H3O) + adduct ion (Xing et al., 2017), in which DP of COS in this study was 3-6. In general, COS, also known as chitosan oligomers or chitooligomers, are made up of chitosan with a degree of polymerization (DP) that is less than 20 (Mourya et al., 2011). Based on DP, our result was consistent with previous study, in which method used for preparing COS from chitosan is appropriate (Fedoseeva et al., 2006). For molecular weight distribution, we measured using gel permeation chromatography (GPC), in which Pullulan (MW: 180–805 000 Da) was used to prepare a calibration curve. This detail has been provided in text as per the reviewer’s suggestion (please see line 82-84, page 2).
For yield, we have to apologize. We did not measure the total yield of COS from chitosan. We focused on characterization, solubility, chemical composition and bioactivities of COS prepared using oxidative hydrolysis method, in which the research had been published in International Journal of Food Science & Technology (Mittral et al., 2022). In this study, we focused on the influence of COS at different levels on viability of LPS-activated RAW 264.7 cells. Also, its ability in inhibiting pro-inflammatory cytokines and the related pathways were also examined. Thank you for comment.
Reference
Fedoseeva EN, Smirnova LA, Sorokina MA, Pastukhov MO.Radicaldegradation of chitosan under the action of a redox system. Russ J ApplChem. 2006. 79:845-849.
Mittal, A., Singh, A., Hong, H., Benjakul, S. Chitooligosaccharides from shrimp shell chitosan prepared using H2O2 or ascorbic acid/H2O2 redox pair hydrolysis: characteristics, antioxidant and antimicrobial activities. International Journal of Food Science & Technology 2022, 58, 2645-2660. doi: 10.1111/ijfs.15696.
Xing, R., Liu, Y.L., Li, K.C. et al. (2017). Monomer composition of chitooligosaccharides obtained by different degradation methods and their effects on immunomodulatory activities. Carbohydrate Polymers, 157, 1288–1297.
Mourya, V. K., Inamdar, N.N., Choudhari, Y. M. Chitooligosaccharides: Synthesis, Characterization and Applications. Polymer Science Series A 2011, 53, 583-612.
Line 74, chitosan was given by Marine Bio Resources Co., Ltd, is it related with Pacific White Shrimp Shell Chitosan in title? Why did you emphasize the Pacific White Shrimp Shell Chitosan? Does COS from others have not the anti-inflammation and antioxidant functions?
******Yes, chitosan was given by Marine Bio Resources Co., Ltd, is it related with Pacific White Shrimp Shell Chitosan in title. Chitosan was prepared from Pacific White Shrimp Shell. Thus, we emphasize the sources (Pacific White Shrimp Shell) in the title. Previous studies showed that COS from other sources also have the anti-inflammation and antioxidant functions (Anil 2022; Jitprasertwong et al., 2021; Synowiecki and Al-Khateeb, 2003). Those information have been added in text (page 2, line 61-63). Thank you for comment.
References
Anil, S. Potential medical applications of Chitooligosaccharides. Polymers 2022, 14(17), 3558. doi: 10.3390/polym14173558.
Synowiecki, J.; Al-Khateeb, N.A. Production, properties, and some new applications of chitin and its derivatives. Crit. Rev. Food Sci. Nutr. 2003, 43, 145–171
Line 73, italic M and subscript w for Mw. Line 80, italic for “g”. Lines 96 and 102, 5 × 104 (not x). Line 108, 35 mm. Line 128, P-value.
*****Sorry for the mistake. We have revised as pre the reviewer’s suggestion. Please see line 73, 102, 107. For ‘g’ and ‘P value’, we removed the content since high similarity index. We need to edit the text following the editor’s suggestion. Sorry for this.
Also, we have attached the revised manuscript in the box. Please see the attached file.
Thank you for your kind suggestion.
